# The Influence of Exercise and Physical Activity on Autonomic Nervous System Function Measured by Heart Rate Variability in Individuals with Type 1 Diabetes Mellitus—A Systematic Review

**DOI:** 10.3390/ijms26157096

**Published:** 2025-07-23

**Authors:** Isabel Bekker, Arne Kooistra, Peter R. van Dijk, Joop D. Lefrandt, Nic J. G. M. Veeger, André P. van Beek

**Affiliations:** 1Diabetes Rehabilitation, Centre for Rehabilitation, University of Groningen, University Medical Centre Groningen, 9713 GZ Groningen, The Netherlands; i.bekker@umcg.nl (I.B.); a.kooistra01@umcg.nl (A.K.); 2Diabetes Center, Department of Internal Medicine, Isala, 8025 AB Zwolle, The Netherlands; p.r.van.dijk@umcg.nl; 3Department of Endocrinology, University of Groningen, University Medical Centre Groningen, 9713 GZ Groningen, The Netherlands; 4Department of Vascular Medicine, University of Groningen, University Medical Centre Groningen, 9713 GZ Groningen, The Netherlands; j.d.lefrandt@umcg.nl; 5Frisius MC Academy, Frisius MC, 8934 AD Leeuwarden, The Netherlands; n.j.g.m.veeger@umcg.nl; 6Department of Epidemiology, University of Groningen, University Medical Centre Groningen, 9713GZ Groningen, The Netherlands

**Keywords:** type 1 diabetes mellitus, autonomic nervous system function, heart rate variability, exercise, physical activity

## Abstract

Non-pharmacological interventions, such as physical activity and exercise, are essential in managing type 1 diabetes mellitus by improving glycemic control, cardiovascular health and autonomic function. Given the chronic nature and long-term complications associated with type 1 diabetes, strategies beyond pharmacotherapy are essential. This review examines the effects of exercise on heart rate variability, a key indicator of autonomic nervous system activity. A systematic search was conducted in March 2024 across PubMed, Embase, Cochrane and CINAHL databases. Studies evaluating the retrospective or prospective impact of exercise or physical activity on heart rate variability parameters were included. Utilizing best evidence synthesis, the methodological quality of the included studies was evaluated. Seven studies met the inclusion criteria, all of which were rated as methodologically weak. Moderate evidence suggests that exercise may enhance heart rate variability, particularly by increasing parasympathetic activity and improving sympathovagal balance. However, evidence remains limited regarding the optimal type, frequency and intensity of exercise. Exercise appears to support autonomic function in individuals with type 1 diabetes mellitus. Nonetheless, further high-quality research is needed to determine the most effective exercise modalities and to inform evidence-based clinical guidelines.

## 1. Introduction

Type 1 diabetes mellitus is a chronic metabolic autoimmune disorder with a rising global incidence, affecting around 8.4 million people worldwide in 2021 [1,2,3]. Effective management of type 1 diabetes mellitus focuses on maintaining optimal glycemic control, primarily through exogenous insulin therapy [3,4]. Poor glycemic control is linked to an increased risk of both microvascular and macrovascular complications [5,6]. Preventing these complications is a key treatment goal and requires a comprehensive approach, which includes, among others, lipid regulation, blood pressure control and body weight management [7]. Beyond pharmacotherapy, non-pharmacological strategies are vital in managing type 1 diabetes mellitus and minimizing diabetes-related complications [8].

Exercise and daily physical activity are important non-pharmacological strategies to improve blood glucose regulation, insulin sensitivity and metabolic control in individuals with type 1 diabetes mellitus [9,10,11]. Physical activity in general, defined as “any bodily movement produced by skeletal muscles that results in energy expenditure” [12], and exercise, comprising planned, structured and repetitive bodily movements aimed at improving or maintaining physical fitness [12], have proven to be beneficial for glycemic control, cardiorespiratory endurance and muscle strength [13]. Moreover, positive effects have also been observed on the autonomic nervous system [14], which can be accurately monitored by heart rate variability (HRV) [15,16]. With HRV, the interval between two heartbeats is measured and analyzed. Variables resulting from this analysis reflect autonomic nervous system activity through sympathetic and parasympathetic involvement and balance and have thus emerged as a key indicator of autonomic nervous system functioning [17]. Time-domain parameters, such as the standard deviation of normal-to-normal (NN) intervals (SDNN) and the root mean sum of squares of differences between adjacent NN intervals (RMSSD), primarily reflect parasympathetic activity. Frequency-domain measures, including low-frequency (LF) and high-frequency (HF) power components, provide additional insight into the balance between sympathetic and parasympathetic modulation [18]. Together, these parameters offer a comprehensive view of autonomic regulation and are frequently used in both research and clinical settings.

Research suggests that exercise leads to improved (i.e., increased) HRV, as observed in healthy individuals [16], elderly [19], children with chronic diseases [20] and in individuals with type 2 diabetes [21]. This improvement is typically reflected in increased parasympathetic related parameters, such as RMSSD and HF power, while markers of sympathetic dominance, such as the LF/HF ratio, may decrease or normalize, indicating a more balanced autonomic regulation. However, the higher glycemic variability in individuals with type 1 diabetes mellitus, particularly during exercise and physical activity, complicates translation from these populations into individuals with type 1 diabetes mellitus. Hyper- and hypoglycemia have been shown to decrease HRV by changing the balance between sympathetic and parasympathetic activity [22,23,24], potentially leading to a complex interaction between physical activity and autonomic functioning [25]. Nevertheless, HRV could be a useful parameter in daily practice, as higher HRV has been associated with decreased mortality [26], improved quality of life [27] and a decreased risk of vascular complications in the long term [28].

This review explores the impact of exercise on autonomic nervous system function, assessed through HRV, in individuals with type 1 diabetes mellitus. By analyzing this relationship, the review provides a comprehensive overview of the potential benefits of exercise, emphasizing its role in improving autonomic function and contributing to overall health and disease management in this population.

## 2. Materials and Methods

This systematic review was conducted in 2024 according to the Preferred Reporting Items for Systematic Reviews and Meta-Analyses (PRISMA) guidelines [29]. The protocol was not registered, as the research was conducted as part of a master’s degree program.

### 2.1. Search Strategy and Selection of Studies

The electronic databases PubMed, Embase, CINAHL and Cochrane were searched for eligible studies in the period from February to the end of March 2024. The search did not have restrictions on the publication date of papers or language. A search string was composed of various MeSH terms and keywords, inter alia “exercise” or “physical activity”, “Diabetes Mellitus, Type 1” and “Heart Rate Variability” or “R-R Interval”. A full search string is provided in the Appendix A. Studies were eligible for inclusion if they (1) included individuals with type 1 diabetes mellitus, (2) included physical activity and/or exercise measured either prospectively or retrospectively and (3) included HRV parameters as outcome measures. Analysis of the RR intervals, or normal-to-normal intervals, resulted in time-domain variables, such as standard deviation of all normal-to-normal intervals (SDNN) or the square root of the mean sum of squares of differences between adjacent NN intervals (RMSSD), and frequency-domain variables, such as high-frequency power (HF) or total power (TP) [18]. Studies were excluded if they (1) included individuals with type 1 diabetes mellitus diagnosed with cardiac autonomic neuropathy and/or cardiovascular disease, as HRV parameters are influenced by these existing comorbidities [30,31], (2) only included exercise testing, (3) only reported on the heart rate or blood pressure measures from which HRV parameters could not be calculated or (4) were pilot studies and cross-sectional studies [30,31].

Two independent reviewers, A. Kooistra and I. Bekker, conducted the selection of eligible studies. The initial screening was based on the title and abstract to assess potential eligibility. Full-text versions were retrieved for studies deemed potentially relevant. Each full-text study was then thoroughly reviewed and assessed against predefined inclusion and exclusion criteria. Furthermore, reference lists of all included studies were systematically examined to identify additional relevant studies through reference tracking.

The reviewers independently extracted data using a standardized extraction form. The data collected from the included studies comprised (1) study characteristics, including first author, publication year and study design; (2) sample demographics; (3) HRV parameters; (4) details of the intervention; and (5) relevant outcomes. In cases where data were incomplete or additional clarification was needed, efforts were made to contact the corresponding author to obtain the missing information.

### 2.2. Assessment of Methodological Quality and Synthesis

Two reviewers (A. Kooistra; I. Bekker) independently assessed the methodological validity of the included studies. The methodological quality was scored using the effective public health practice project (EPHPP) assessment tool for quantitative studies developed by McMaster University [32]. The tool was chosen, as it was developed to encompass a broad variety of research designs [32]. This tool consists of eight components assessing methodological quality based on selection bias, study design, confounders, blinding, data collection methods, withdrawals and drop-outs, intervention integrity and analyses. However, for the purpose of the global score, only selection bias, study design, confounders, blinding, data collection methods, withdrawals and drop-outs were considered. Intervention integrity and analyses were not factored into the global score. Each component is scored either as strong (+), moderate (~), weak (-) or not applicable (x), based on the available information provided by the authors of the study. The global score is determined based on the frequency and severity of each individual component rating. The study was considered to have strong methodological quality with no “weak” ratings, moderate quality with only one “weak” rating and weak quality with two or more “weak” scores.

A narrative best evidence synthesis was performed for all included studies, in which the strength of evidence was determined using a classification for assessment adapted from Proper et al. [33]. The results were considered to be consistent when at least 75% of the studies showed results in the same direction, defined according to significance (*p* ≤ 0.05). The term risk of bias, as used by Proper et al. [33], was replaced with the term methodological quality to ensure language consistency. Three levels of evidence were distinguished. Strong evidence was considered to be consistent findings in multiple (>2) high methodological quality studies. Moderate evidence was considered to be consistent findings in one high or moderate methodological quality study and at least one weak methodological study or consistent findings in multiple high risk of bias studies. Lastly, insufficient evidence was considered in cases where only one study was available or for inconsistent findings in multiple studies.

## 3. Results

### 3.1. Study Selection

The initial search strategy yielded 119 results, of which 45 were identified as duplicates and excluded. The remaining studies were screened based on their titles and abstracts, resulting in 25 studies selected for full-text retrieval. Full-text access could not be obtained for seven of these studies.

Eighteen studies received full-text screening. Two studies were excluded based on population, as one only included individuals with type 2 diabetes mellitus [34], while the other one included individuals diagnosed with cardiac autonomic neuropathy [35]. One study was excluded due to the fact that HRV parameters were only analyzed in relation to HbA1c concentration [36]. Furthermore, one study [37] including both individuals with type 1 diabetes mellitus and type 2 diabetes mellitus was excluded because only between-group analysis was performed, and no within-group parameters were available. Another study was excluded for only including heart rate changes, and HRV parameters could not be calculated from the available data [38]. One study only included exercise testing and no exercise or physical activity parameters [39]. The remaining six studies were excluded based on their study design, including one pilot study [40] and five cross-sectional studies [25,30,40,41,42,43]. Additionally, one study was added after screening of references [44], finally leading to seven included studies [44,45,46,47,48,49,50]. An overview of the selection process according to PRISMA guidelines [29] is displayed in Figure 1.

### 3.2. Study Characteristics

In Table 1, an overview of study characteristics, sample information, HRV parameters, intervention and relevant results are given. Despite searching for studies across all age groups, six of the seven studies included children as individuals in their studies [44,45,46,47,48,50]. Of these six, one study included both children and adults in its sample [46], whereas one study included only adults [49]. Three of the included studies described an exercise intervention [48,49,50]; one study described non-mandatory sports activities during a summer camp [45]; and one study described one acute exercise session on a stair stepper [36]. Three studies included self-reported physical activity measures [44,45,46,47], whereas one study measured physical activity with a triaxial accelerometer [46]. The intensity of the intervention was described in three [48,49,50] of the seven studies. Five studies had a control group with between-group analysis. Three studies had healthy controls [46,47,48], and three studies had individuals with type 1 diabetes mellitus as controls [45,48,49]. One study [50] only performed within-group analysis, and one study [44] subdivided the included individuals according to the amount of physical activity and performed within- and between-group analyses. For the measurement of HRV parameters, five studies used an electrocardiogram (ECG) [45,46,47,48,50], a reliable and valid measurement method [51]. Two studies used heart rate monitors [45,49] to gain insight into HRV parameters, and one study used a telemetry system [44].

### 3.3. Methodological Quality

The results of the methodological quality assessment are shown in Table 2. No study included blinding of either individuals, researchers or analysts in their design. None of the included studies provided explicit reporting of their study designs; thus, the determination of study designs was made by the reviewers. Six studies reported [44,45,46,47,49,50] on the management of confounders either by design or analysis. The included studies were all categorized as having a weak methodological quality.

### 3.4. Results of Individual Studies

All of the included studies reported analyses on frequency-domain variables, and four studies included time-domain variable analyses [45,46,48,49]. In all of the selected studies, a *p*-value of <0.05 was considered significant.

Following the exercise intervention, pre/post analyses of HRV parameters were conducted in five studies. These analyses showed significant improvements in parasympathetic modulation measured through the root mean square of successive differences between normal heartbeats (RMSSD) [45,48,49] and high-frequency (HF) power [45,48,50]. Low-frequency power (LF) [48,50] as a measure of sympathetic and parasympathetic activity and total power (TP) [50] as a reflection of total autonomic nervous system function improved as well. Furthermore, significant decreases in low- and high-frequency ratio (LF/HF) [48,49] and very-low-frequency and high-frequency ratio (VLF/HF) [45], reflecting sympathovagal balance, were reported. One study reported significant increases in the standard deviation of all NN intervals (SDNN) [48]. One study reported no significant differences in pre/post analyses in absolute values in RR interval, LF [44], and two studies reported no significant changes in HF [44,50].

Three studies analyzed the results compared to a control group [45,47,48], of which one study [47] found no significant difference in the active state between individuals with type 1 diabetes mellitus and healthy controls. Another study [48], on the other hand, found significant improvements in the intervention group with individuals with type 1 diabetes mellitus compared to a control group with individuals with type 1 diabetes mellitus.

In the resting state, individuals with type 1 diabetes mellitus, compared with healthy controls or untrained individuals with type 1 diabetes mellitus, reported significantly lower HF and LF [45,47]. One study [48] found significantly higher HRV parameters in the healthy control group at baseline compared to individuals with type 1 diabetes mellitus.

Furthermore, associations were found between the amount of physical activity and HRV parameters in four of the included studies [44,45,47]. Two studies reported that untrained or less physically active individuals had a higher occurrence of abnormal HRV parameters [45,47]. The remaining two studies reported a decrease in RMSSD, HF and LF with less physical activity or increased sedentary time [44,46]. The results are displayed in Table 1.

### 3.5. Synthesis of Evidence

Based on the EPHPP scores, all studies were considered to be of weak quality [44,45,46,47,48,49,50]. Five studies measuring the influence by means of within-group analyses [44,45,46,47,50] were included for best evidence synthesis. Due to the heterogenous control groups, between-group analyses were not included in the best evidence synthesis. The study by Marshall et al. [46] was excluded from best evidence synthesis because neither within- nor between-group analyses were reported.

Overall, five [44,45,48,49,50] of the six studies found a positive significant within-group effect of exercise or physical activity on HRV parameters, whereas one study [47] reported a negative significant effect of exercise or physical activity on HRV parameters. Despite the weak methodological quality of the included studies, the consistency in findings suggests moderate evidence of a positive influence of exercise or physical activity on HRV parameters, and thus, autonomic nervous system function.

A sub-synthesis was performed to investigate the influence of exercise or physical activity on autonomic nervous system function through either sympathovagal balance, parasympathetic or sympathetic activity.

For the sub-synthesis on sympathovagal balance, six studies [44,45,47,48,49,50] were included for the synthesis. Five [44,45,48,49,50] of the six studies reported a positive significant effect of exercise or physical activity on sympathovagal HRV parameters, indicating moderate evidence of a positive influence of exercise and physical activity on sympathovagal balance.

Five studies were included for the sub-synthesis of parasympathetic activity. One study [50] did not report statistically significant results for parasympathetic HRV measures and was thus excluded from the synthesis. Four [44,45,48,49] of the five studies reported a significant positive effect on parasympathetic activity, whereas one study [47] reported a significant decrease in parasympathetic activity in children. This indicates moderate evidence of a positive influence of exercise or physical activity on parasympathetic activity.

Three studies [45,48,50] reported on sympathetic activity, with two studies reporting an increase in sympathetic activity and one study a decrease. These inconsistent findings result in insufficient evidence to support either a positive or negative effect of exercise or physical activity on sympathetic activity.

Sub-synthesis evidence of the type, frequency or intensity of the interventions in the included studies is insufficient due to the heterogeneity of the interventions of the included studies.

## 4. Discussion

This review aimed to summarize the available evidence on the influence of exercise and physical activity on the autonomic nervous system, as measured by HRV parameters, in individuals with type 1 diabetes mellitus. While the findings cautiously suggest that exercise may have a beneficial effect on parasympathetic activity and the balance between sympathetic and parasympathetic activity, evidence of direct changes in sympathetic activity remains limited. Importantly, this review focused on the general impact of exercise rather than evaluating the effectiveness of specific intervention types. However, variability in the type, frequency and intensity of the interventions across studies led to inconsistent findings, highlighting the need for further research to establish clear recommendations. Nevertheless, these results reinforce the potential of exercise to enhance autonomic function in individuals with type 1 diabetes mellitus.

The observed positive influence of exercise on parasympathetic activity and sympathovagal balance can be attributed to several mechanisms. Acute exercise activates central command and the exercise pressor reflex [52,53], which increase sympathetic outflow and reduce parasympathetic activity to meet the metabolic demands of active muscles [54,55]. Simultaneously, the arterial baroreflex is reset to operate at a higher blood pressure range, allowing for appropriate cardiovascular responses [55,56]. With regular training, these acute effects lead to chronic adaptations, including enhanced vagal tone and baroreflex sensitivity, reflecting neuroplastic changes within autonomic control centers [57]. These adaptations may contribute to the improved heart rate variability. This aligns with evidence from other studies showing increased activity of vagal preganglionic neurons and neuroplastic changes in key autonomic brain regions [57,58]. These adaptations contribute to better autonomic balance and preservation of parasympathetic tone, even in pathological conditions [58]. However, the evidence on sympathetic activity remains inconclusive, with some studies showing reductions in sympathetic nerve activity and others failing to demonstrate significant effects [54]. This inconsistency may stem from differences in study designs, participant characteristics and methods for assessing sympathetic function.

Support for these findings comes from systematic reviews in other populations. Estevez et al. [20] reported that exercise programs improved sympathovagal balance in children with chronic diseases, although their results primarily involved time-domain parameters, contrasting with the frequency-domain improvements observed in this review. Picard et al. [21] demonstrated robust evidence for exercise-induced improvements in HRV in individuals with type 2 diabetes mellitus, particularly in frequency-domain parameters, and identified evidence regarding the type, intensity and frequency of exercise. Similarly, Hamasaki et al. [59] found significant enhancements in autonomic function through exercise, focusing on heart rate recovery and baroreflex sensitivity. Differences in the populations and methodologies used in these reviews may account for variations in findings.

The review’s limitations include the lack of analysis on glycemic control and body composition, both of which significantly influence HRV in individuals with type 1 diabetes mellitus. Improved glycemic control through physical activity increases insulin sensitivity but also heightens the risk of hypoglycemia, which can affect autonomic responses [60,61]. Also, body composition significantly impacts HRV parameters [62,63,64]. For instance, increased adiposity is associated with heightened sympathetic activity and reduced parasympathetic activity, likely due to chronic low-grade inflammation and metabolic dysregulation [65]. These mechanisms emphasize the importance of considering body composition in understanding HRV. However, the absence of obese individuals in the included studies limits the generalizability of the results to overweight or obese individuals with type 1 diabetes mellitus. Additionally, sex-specific differences in HRV, as reported by Koenig et al. [66], were not addressed in the majority of studies. Their findings suggest that women exhibit greater parasympathetic modulation of HRV, while men demonstrate a predominance of sympathetic activity. These distinctions underscore the necessity for future research to account for sex differences in HRV analyses.

Finally, most studies focused on children and adolescents [67], whose developmental stages and hormonal changes could influence autonomic function, making it challenging to generalize the findings to adults [68].

The limitations of the review process itself included the heterogeneity of study designs, interventions and outcomes, which precluded meta-analysis and limited the ability to draw definitive conclusions.

Publication bias was not addressed in the review process. Publication bias refers to the possibility of bias due to publication or non-publication of research findings depending on the results. Findings with positive results are more likely to be published [69]. As the research protocol was not published, and no analysis was included to detect heterogeneity of the studies or address publication bias, it is unclear what influence potential publication bias had on this review’s results [70].

The methodological quality of the included studies was generally low, as assessed using the EPHPP tool. While the EPHPP is a well-established instrument for evaluating public health research, it is not specifically designed to assess exercise-based interventions. As a result, certain aspects relevant to exercise studies may not have been adequately captured. Furthermore, the clinical relevance of changes in HRV remains uncertain. This lack of quantification limits the practical application of these findings and complicates translation into exercise recommendations or clinical monitoring strategies. In addition, it remains unclear which type, intensity or duration of exercise is most beneficial. For future research, we recommend using assessment tools specifically designed for exercise interventions to enhance methodological rigor. Studies should aim for greater homogeneity in intervention protocols and outcome measures to allow for synthesis and comparison. Furthermore, future investigations should quantify the magnitude and clinical relevance of changes in HRV, including establishing minimal clinically important differences. Additionally, studies should address glycemic control, body composition and sex differences. Including diverse populations, particularly adults and those with varying body compositions, will enhance the applicability of the findings to clinical practice.

The implications for practice and policy suggest that HRV may be a valuable, non-invasive tool for assessing autonomic function, particularly parasympathetic activity, although its utility for evaluating sympathetic activity remains less well established. Wearable technology offers the potential for real-time monitoring of autonomic responses, which could support individualized care. Clinicians can consider integrating physical activity guidelines into the management of individuals with type 1 diabetes mellitus, highlighting its potential role in autonomic regulation alongside glycemic control and cardiovascular health. Future research could further refine heart rate variability metrics, investigate its relevance as a biomarker for autonomic health and explore its potential application in developing personalized exercise recommendations for managing type 1 diabetes mellitus. Regular physical activity remains an essential non-pharmacological strategy for this population.

## 5. Conclusions

The findings indicate, with some reservation, that exercise may play a valuable role in supporting autonomic function in individuals with type 1 diabetes mellitus. However, the current evidence is constrained by the low methodological quality of the included studies and the lack of consistent data on the effects of different exercise modalities, intensities and frequencies. High-quality experimental research is urgently needed to clarify these effects and provide evidence-based clinical recommendations. Future studies should also consider factors such as glycemic control, body composition and sex-specific differences to enhance the relevance and applicability of the findings, enabling tailored exercise interventions for this population.

## Figures and Tables

**Figure 1 ijms-26-07096-f001:**
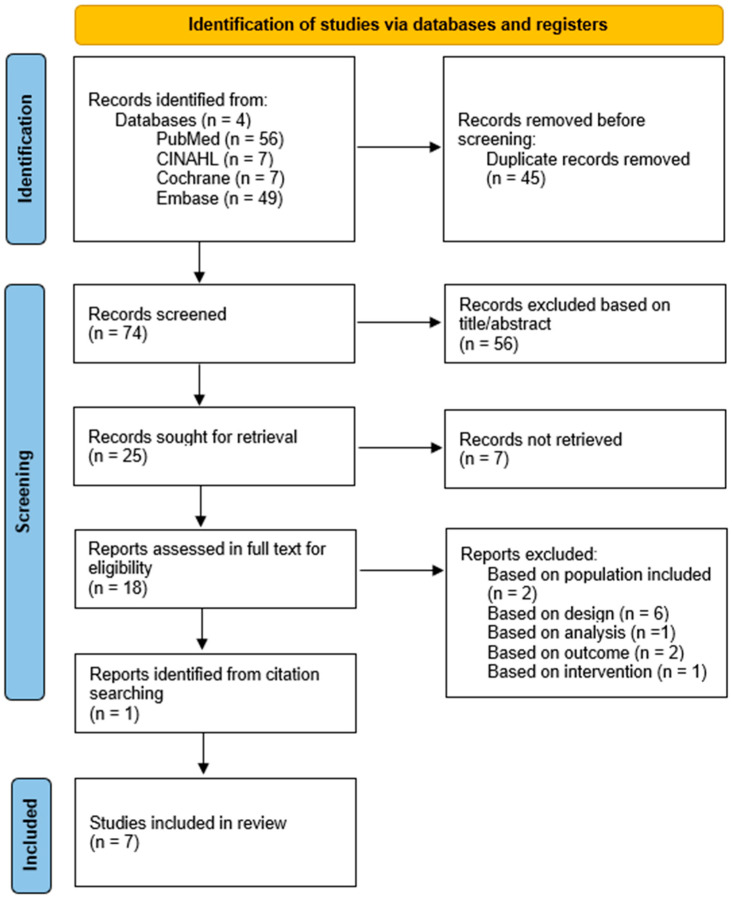
PRISMA flowchart (2020) of included studies.

**Table 1 ijms-26-07096-t001:** Overview of study characteristics, sample information, HRV parameters, intervention and relevant results.

Study Characteristics		Sample			HRV Parameters		Intervention ^§^	Relevant Results
Reference	Design ^†^	N total (m/f)	Type	Age ± SD	Method	Variables		
Alarcón-Gomez et al. 2021 [49]	Randomized experimental, parallel, open-label trial	19 (10/9)	I: T1DM C: T1DM	I:38 ± 5.5 C:35 ± 8.2	Heart Rate Sensor (Polar H10 with Polar Pro strap)	RMSSD LF/HF	Exercise intervention: HIIT training 30 s peak at 85% of PPO 1 min recovery at 40% PPO Duration: 3× per week for 6 weeks C: No exercise	I: RMSSD pre/post conditions 37.8 ± 27.9 vs. 44.3 ± 27.7 (*p* < 0.05, ES: 0.22) LF/HF pre/post conditions 2.6 ± 1.6 vs. 1.5 ± 0.9 (*p* < 0.05, ES: 0.23) Significant interaction condition × time in LF/HF (*p* < 0.05) C: No significant changes in control group RMSSD pre/post conditions 40.0 ± 15.9 vs. 39.3 ± 16.6 LF/HF ratio pre/post conditions 2.1 ± 2.0 vs. 1.9 ± 2.2
Saki et al. 2023 [48]	N/R Clinical controlled trial	36 (36/0)	I: T1DM C1: T1DM C2: Healthy	I: 15.6 ± 1.80 C1: 15.25 ± 1.76 C2: 15.08 ± 1.67	ECG Holter monitor	SDNN RMSSD HF LF LF/HF	Exercise intervention: HIIT running and swimming 10–15 min w/u, stretching, training and 10 min c/u Running: 30–55 min with 3–6 periods of 5 min at 50–75% HRR and 4 min recovery at 10–20% HRR. Swimming: 10–15 mini front crawl leg, 30–55 min with 3–6 periods of 5 min full front crawl at 50–75% and 4 min recovery at 10–20% of HRR. Resistance training Duration: 3× per week for 12 weeks	Pre-test HRV was significantly different in C_2_ group compared to I and C_1_ group. Significant ME in the intervention group for SDNN, RMSSD, LF, HF and LF/HF after intervention. Significant between-group differences after intervention (adjusted means ± SE): I vs. C_1_: SDNN 137.19 ± 3.00 vs. 135.02 ± 2.33 (*p* < 0.001) RMSSD 41.10 ± 1.68 vs. 40.86 ± 1.38 (*p* < 0.001) LF 1465.23 ± 40.26 vs. 1544.64 ± 35.14 (*p* < 0.001), HF 737.71 ± 45.50 vs. 779.56 ± 29.59 (*p* < 0.001) LF/HF 2.15 ± 0.05 vs. 2.24 ± 0.03 (*p* < 0.001) I group improved significantly better than C_1_ group. I vs. C_2_: SDNN 137.19 ± 3.00 vs. 156.34 ± 2.23 (*p* < 0.001) RMSSD 41.10 ± 1.68 vs. 48.38 ± 1.22 (*p* < 0.001) LF 1465.23 ± 40.26 vs. 2042.48 ± 44.26 (*p* < 0.001) HF 737.71 ± 45.50 vs. 1312.28 ± 29.29 (*p* < 0.001) LF/HF 2.15 ± 0.05 vs. 1.37 ± 0.04 (*p* < 0.001) C_2_ group was significantly better than I group. No significant differences between C_1_ and C_2_ after testing
Chen et al. 2007 [47]	Retrospective pre/post design	200 (95/105)	I: T1DM C: Healthy	I:10.3 ± 1.6 C:10.4 ± 1.6	Three-channel ECG	LF HF LF/HF	I: PAQ-C, in which children were divided into Low (<2), Moderate (>2 and <3) or High (>3) activity levels based on the past 7 days. Stair stepper for 10 min C: Same as I	Significant between-group differences: LnHF (4.9 ± 0.9), LnLF (6.0 ± 0.7) and LnTP (6.7 ± 0.4) significantly ↓ in resting state (*p* < 0.05). No significant differences in LnHF/LF. In active state, no significant differences between T1DM and controls: LnHF (2.2 ± 1.1 vs. 2.3 ± 1.1), LnLF (4.0 ± 0.9 vs. 4.1 ± 0.8), LnHF/LF (1.7 ± 0.5 vs. 1.7 ± 0.7) and LnTP (5.0 ± 0.8 vs. 5.0 ± 0.7) Significantly decreased within-group changes when going from resting to active state in LnHF, LnLF and LnTP. PA predicted 54% of the variance in HRV parameters (*r* = −0.21, *p* < 0.05).
Javorka et al. 2001 [45]	N/R Prospective cohort study	20 (N/R)	I: Trained T1DM C: Non-trained T1DM	15.5 ± 1.2	ECG signal through chest belt (Varia Pulse TF3 System)	RR MSSD HF LF VLF TP LF/HF VLF/LF LF/HF	Reconditioning summer camp for 8 days including non-mandatory activities: table tennis, badminton, 3–5 km distance walking, hiking, swimming 2× per day. Self-reported weekly physical activity levels for the last 6 months. C: Same as I	Trained subgroup compared to non-trained group had significantly longer RR intervals, ↑MSSD, VLF, HF and LF↑ (*p* < 0.05). Abnormal HRV parameters in 14.6 ± 6% of trained individuals vs. 54 ± 9% in untrained. Significant changes before and after with n total. No relevant differences between trained and untrained group (*p* < 0.05): RR↑ (715 ± 23 vs. 786 ± 22) MSDD↑ (2103 ± 714 vs. 6072 ± 1603) HF↑ (1024 ± 411 vs. 2195 ± 547) Rel.P.VLF↓ (47.6 ± 3.6 vs. 33.5 ± 4.1) Rel.P.HF↑ (30.4 ± 4.1 vs. 47.9 ± 5.1) VLF/HF↓ (2.8 ± 0.6 vs. 1.1 ± 0.3)
Lucini et al. 2012 [44]	N/R Prospective cohort study	77 (50/27)	T1DM	15.0 ± 0.6	Two-way radio telemetry system (Finapres)	RR VAR_RR_ LF_RR_ HF_RR_ LF/HF_RR_ Units used: ms and nu	METS calculation through assessment of time spent walking (>10 min) and/or exercise (structured or leisure time). At T1, subgroups were designated according to increased (Group 1), unchanged (Group 2) or diminished (Group 3) total weekly METS.	Significant interaction of between-group results in RR (*p* = 0.04), LF_RR_ [nu] (*p* = 0.03) and HF_RR_ [nu] (*p* = 0.01). Group 1: RR↑ (835. 1 ± 24.7 vs. 885.6 ± 23.2), VAR↑ (4443 ± 703 vs. 5042 ± 755), LF(ms)↓ (1176 ± 220 vs. 1152 ± 170), LF(nu)↓ (44.2 ± 3.2 vs. 41.1 ± 3.2) HF(ms)↑ (1918 ± 436 vs. 2425 ± 533), HF(nu)↑ (47.8 ± 2.9 vs. 52.6 ± 3.0) LF/HF↓(1.5 ± 0.3 vs. 1.1 ± 0.2) Exercise amount ↑ (METS/p.m/p.w.) is associated with a small reduction in LF_RR_ and increase in HF_RR._
Marshall et al. 2021 [46]	N/R Prospective cohort study	37 (21/16)	I: T1DM C: Non-diabetic subjects	I: 11.9 ± 1.5 C:11.6 ± 2.2	ECG	RMSSD LF HF	Triaxial accelerometer measurement for 28 days, 24 h p.d. SED, LPA, MVPA and sleep time were determined. C: Same as I	RMSSD is significantly negatively associated with SED (γ = −28.94, *p* < 005) and non-significantly negatively associated with MVPA (γ = −3.06, *p* > 0.05). Positive association with LPA (γ = 18.13) and sleep (γ = 13.87), both non-significant (*p* > 0.05). LF was negatively associated with SED (γ = −2.25), MVPA (γ = −1.76) and LPA (γ = −8.09) (non-significant). Positive association with sleep (γ = 12.10) HF was only negatively associated with sleep (γ = −12.04) but positively associated with SED (γ = 2.24), LPA (γ = 8.05) and MVPA (γ = 1.75) (non-significant).
Shin et al. 2014 [50]	N/R Prospective cohort study	15 (15/0)	T1DM	13.0 ± 1.0	ECG monitor	TP VLF LF HF SNS index	Walking exercise program 3× p.w. for 12 weeks. Exercise intensity was set at 60% of VO2max. Duration requirement was at least 250 kcal per exercise session.	Pre/post HRV power analysis parameters all increased significantly (*p* < 0.05): TP (998.46 ± 232.2 vs. 1587.47 ± 449.25) LF (541.26 ± 187.59 vs. 942.22 ± 397.84) VLF (128.11 ± 58.66 vs. 389.44 ± 198.55) SNS index and HF power were not significantly different before and after testing.

I: Intervention group, C: Control group, N/R: Not reported. ^†^ When design was not reported, I. Bekker determined the study design. All HRV parameters and measurement units are explained in Appendix B. Ln(HRV): Logarithmic transformation for analysis, Rel.P: Relative power, ms: Milliseconds. ^§^ HIIT: High-intensity interval training, PPO: Peak power output, PAQ(-C): Physical activity questionnaire (children), PA: Physical activity, METS: Metabolic equivalents, SED: Sedentary, LPA: Light physical activity, MVPA: Moderate-to-vigorous physical activity, VO_2_max: Maximum oxygen consumption, vs.: Versus, p.w.: Per week, p.m.: Per month, p.d.: Per day.

**Table 2 ijms-26-07096-t002:** Risk of bias assessment.

Study	1	2	3	4	5	6	Global Rating
Alarcón-Gómez et al. [49]	~	−	+	−	+	~	Weak
Chen et al. [47]	~	−	+	−	+	−	Weak
Javorka et al. [45]	−	−	+	−	−	−	Weak
Lucini et al. [44]	~	−	−	−	+	−	Weak
Marshall et al. [46]	~	−	+	−	+	~	Weak
Saki et al. [48]	−	−	+	~	+	−	Weak
Shin et al. [50]	−	−	+	−	+	−	Weak

1: Selection bias; 2: Study design; 3: Confounders; 4: Blinding; 5: Data collection method; 6: Withdrawals and drop-outs.

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
