# Peer review of "The Influence of Exercise and Physical Activity on Autonomic Nervous System Function Measured by Heart Rate Variability in Individuals with Type 1 Diabetes Mellitus—A Systematic Review"

_ijms, 2025, doi:10.3390/ijms26157096_

Round 1

Reviewer 1 Report

Comments and Suggestions for Authors

The article is devoted to a relevant topic of modern medicine, namely to evaluation of influence of exercise and physical activity on autonomic nervous system function in Individuals with type 1 diabetes mellitus. Authors used best evidence synthesis in their systematic review to highlight, that physical activity has some beneficial role in individuals with type 1 diabetes mellitus. However, evidenceshown by authors is constrained by the low methodological quality of included studies , which authors highlight as limitation of their study.

In general the manuscript clear, relevant for the field and presented in a well-structured manner. Cited references mostly recent publications (within the last 5 years) and relevant and does not include an excessive number of self-citations. The manuscript is scientifically sound and the experimental design is appropriate to test the hypothesis. The manuscript’s results can be reproduced based on the details given in the methods section.  The review is clear, comprehensive and relevant to the field. The gap in knowledge identified in "Introduction" section. The figure and two tables properly show the data, obtained during screening and best evidence synthesis. They are easy to interpret and understand.  The conclusions are consistent with the evidence and arguments presented in the article. 

Regarding iThenticate report, that showed 28% of similarity. Most of highlighted fragments represent general phrases, typical to scientific style of writing and cannot be considered plagiarism. Maximal percentage of similarity with a single source was 2%, therefore, 28% of similarity shown by iThenticate report cannot be considered a significant drawback of this manuscript. It can be removed by a literature editing. 

Author Response

Dear reviewer, 

Thank you very much for taking time to review our manuscript. Please find a detailed respons to the comments and suggestions rased in the attached file. 

We hope these revisions and clarifications address the concerns raised and demonstrate our commitment to improving the quality and clarity of our work. We thank you once again for your valuable input.

Yours sincerely, 

Arne Kooistra, Universitair Medisch Centrum Groningen, Centrum voor Revalidatie, Locatie Beatrixoord CD33, Beatrixoord Diabetesrevalidatie, 9750WX Haren Gn

Reviewer 2 Report

Comments and Suggestions for Authors

The Influence of Exercise and Physical Activity on Autonomic Nervous System Function Measured by Heart Rate Variability in Individuals With Type 1 Diabetes Mellitus – A Systematic Review

This review aimed to summarize available evidence on the influence of exercise and
physical activity on the autonomic nervous system, as measured by HRV parameters, in
individuals with type 1 diabetes mellitus. Theme is interesting. There are similar articles (for instance: Chen SR, Lee YJ, Chiu HW, Jeng C. Impact of physical activity on heart rate variability in children with type 1 diabetes. Childs Nerv Syst. 2008 Jun;24(6):741-7. doi: 10.1007/s00381-007-0499-y. Epub 2007 Sep 28. PMID: 17901961).

The article was reported according to PRISMA. Is the research protocol available at INPLASY–International Platform of Registered Systematic Review and Meta-analysis Protocols https://inplasy.com/ (accessed on which day?) (please registration number please and DOI number )? Search strategy and selection of the studies were made rigourously. Study selection and characteristics were presented in table 1. Figure 1 illustrates the PRISMA Flowchart (2020) of included studies. References must be up-dated.

Author Response

(The authors gave the same response as above.)

Reviewer 3 Report

Comments and Suggestions for Authors

The authors conducted a systematic review to examine the potential of exercise to improve autonomic function in individuals with type 1 diabetes. The review was certainly rigorous, as it highlighted all the weaknesses of the studies conducted, primarily the evidence on sympathetic activity, which remains inconclusive, with some studies showing reductions in sympathetic nerve activity and others failing to demonstrate significant effects. The authors themselves acknowledge the limitations of the review, including the lack of analyses of glycemic control and body composition, both of which significantly influence HRV in individuals with type 1 diabetes, as well as the failure to consider gender differences in most studies. The authors focused on the limitations of the review process, namely the heterogeneity and methodological quality of the studies, study designs, interventions, and outcomes, which limited the ability to draw definitive conclusions from the meta-analysis, limiting them to indicating, with some reservations, the role that physical exercise may play in supporting autonomic function in individuals with type 1 diabetes mellitus. Based on these observations, the authors provide suggestions regarding the use of assessment tools specifically designed for exercise-based interventions, in order to improve methodological rigor and provide more robust evidence-based clinical recommendations.

Author Response

Dear reviewer, 

Thank you very much for taking time to review our manuscript. Please find a detailed respons to the comments in the attached file. 

We thank you once again for your valuable input and taking the time to thoroughly assess our manuscript. 

Yours sincerely, 

Arne Kooistra, Universitair Medisch Centrum Groningen, Centrum voor Revalidatie, Locatie Beatrixoord CD33, Beatrixoord Diabetesrevalidatie, 9750WX Haren Gn
